# The NIH BRAIN Initiative's impacts in systems and computational neuroscience and team-scale research 2014–2023

Farah Bader[†], Clayton Bingham[†], Karen K David[‡], Hermon Gebrehiwet[§], Crystal L Lantz[§], Grace CY Peng[‡], Mauricio Rangel-Gomez[‡], James Gnadt*[‡], On behalf of the NIH BRAIN Initiative Integrative and Quantitative Neuroscience Team

National Institutes of Health, Bethesda, United States

*For correspondence:
jwgnadt@yahoo.com

[†]Program Analyst

[‡]Program Officer, BRAIN Team Co-Leads (current and emeritus)

[§]Program Officer

Competing interest: The authors declare that no competing interests exist.

## eLife Assessment

The authors provide a **convincing** summary of ten years of Brain Initiative funding including the historical development, the specific funding mechanisms, and examples of grants funded and work produced. It is particularly **valuable** at this moment in history, given the cataclysmic changes in the US government structure and function occurring in early 2025.

**Abstract** At the 10-year anniversary of the NIH BRAIN Initiative, this report analyzes the impact of the initiative's functional neuroscience ecosystem as funding experiments in the domains of systems and integrative neuroscience, and computational neuroscience, with an eye on comparison with other funding models and best practices.

## Introduction

The Brain Research Through Advancing Innovative Neurotechnologies Initiative (BRAIN Initiative) was launched in fiscal year 2014 from a 2013 U.S. Presidential directive, guided by evaluation from an advisory committee to the NIH Director. The advisory committee conveyed to NIH a mission concept in the Brain 2025: A scientific Vision with seven priority areas: (1) Discovering Diversity, (2). Maps at Multiple Scales, (3) The Brain in Action, (4) Demonstrating Causality, (5) Identifying Fundamental Principles, (6) Advance Human Neuroscience, (7) From the BRAIN Initiative to the Brain. A special Congressional appropriation for the BRAIN Initiative provided funds to execute these priorities and allowed the BRAIN Initiative to focus on progressive and adventurous projects and programs outside of standard NIH Institute/Center (IC) constraints, and across IC missions.

The core philosophy of the BRAIN Initiative is to understand the brain as a complex system that gives rise to the diversity of functions that allow us to interact with, and adapt to, our physical and social environments. Such an understanding is necessary to promote brain health and to prevent and treat neurobehavioral and neurological disorders (*Miller et al., 2024*). Like any complex system, one must understand how the brain works in order to know how to fix it when it dysfunctions. All brain disorders, including neurological, mental, and behavioral conditions, manifest as system dysfunctions that must be understood as such for effective therapeutic cures and prevention. Thus, the NIH BRAIN Initiative has created a unique synergy with its partner NIH ICs that dramatically advances basic research on brain function, complementing the mission interests of the ICs in preventing and treating specific diseases.

Under the guidance of the Directors of 10 NIH 'neuro' ICs, Scientific Program Directors and Specialists with expertise in various approaches that matched the advisory committee guidance convened to interpret and implement the mission concepts of the BRAIN 2025 report. The systems neuroscience program– after splitting from the technology development component in FY2015 – was charged with implementing the investigative, functional neuroscience of the mission directives, spanning all seven priority areas, as noted above. Noninvasive approaches in functional human neuroscience were pursued as a separate track focused on neuroimaging technologies across scales.

The Integrative and Quantitative Approaches Team was convened to design funding programs that would address mechanisms of how the nervous system works as an integrated system. Strategically, this fundamental, systems neuroscience funding component focused on combining in vivo behavior, dynamic neural systems, and computational neuroscience approaches (BRAIN 2025), an operational definition of systems neuroscience that overlaps substantially with a newly proposed concept of 'heksor' ("*… a widely distributed network of neurons and synapses that produces an adaptive behavior and changes itself as needed in order to maintain the key features of the behavior, the attributes that make the behavior satisfactory*") (*Wolpaw and Kamesar, 2022*). Since most functional systems of the nervous system are mediated by dynamic neural circuits, this component of the BRAIN Initiative is sometimes referred to as the 'BRAIN Circuits Program'. However, flexibility was built into the funding programs to encourage understanding of non-neuronal systems' influence on dynamic circuits and to allow studies of in vivo 'behavior' of well-defined neural systems within the brain.

## Results

### The NIH BRAIN Circuit Programs

From the beginning, this functional neuroscience approach adopted a set of guiding scientific principles that defined the priorities and context for developing the content of the 'circuit-busting' funding announcements (*David et al., 2020*). Based on these principles, the BRAIN Circuits Program created an ecosystem of funding mechanisms illustrated in *Figure 1A*. The general format of this integrated approaches program has been to have 2/3-year exploratory forms of funding, followed by a peer-reviewed opportunity to compete for an expanded 5-year award, in each emphasis area. Note that the exploratory projects are not a requirement of the expanded awards but offer an enabling, or pilot, step where useful. Research topics were prioritized to be investigator-initiated concepts in fundamental systems neuroscience in three major research tracks: (1) targeted, single-lab-sized or limited multi-PI research proposals, (2) team-research approaches that could only be successful as integrated approaches across biological scales, disciplines, and/or species, (3) human-neuroscience research opportunities afforded by direct, intracranial access to recording and manipulating in the human brain. This format emphasizes peer-reviewed, investigator-initiated selection of scientific merit.

Awards in these BRAIN Circuits Programs were required to include multi-scale approaches combining in vivo behavior, dynamic neural systems, and quantitative methods. *Supplementary file 1* shows a word cloud made of the unique, competing BRAIN Circuits Program award titles and abstracts which illustrates that the salient research topics from this PI-initiated program reflect the research emphases in fundamental behavioral, systems, and computational neuroscience. The spectrum of specific research topics can be mined from the award links provided below. The funding mechanisms, number of awards, and total committed budget through 2023, per funding announcement (Request for Application [RFA]), are listed in *Figure 1B*. Description of funding mechanisms in this table and discussed in this report can be found on the NIH Grants & Funding website. This total 10-year expenditure in quantitative systems neuroscience has invested approximately $1B into fundamental neuroscience research. Note that, as directed by the BRAIN 2025 report, the BRAIN Circuits Program balanced funding between small-lab and collective team-research in the Targeted and Team track at $384M vs. $403M, respectively. The human Research Opportunities in Humans (ROH) program composed about 10% of the total budget at $100M.

As part of the functional neuroscience approach, there was also a strong desire to promote better developed theories into fundamental systems neuroscience. Taking into account that all the BRAIN Circuits Program funding mechanisms were required to include quantitative methods and to not replicate or compete with the highly successful NSF/NIH Collaborative Research in Computational Neuroscience, we crafted a 'tool building' program in computational neuroscience with an

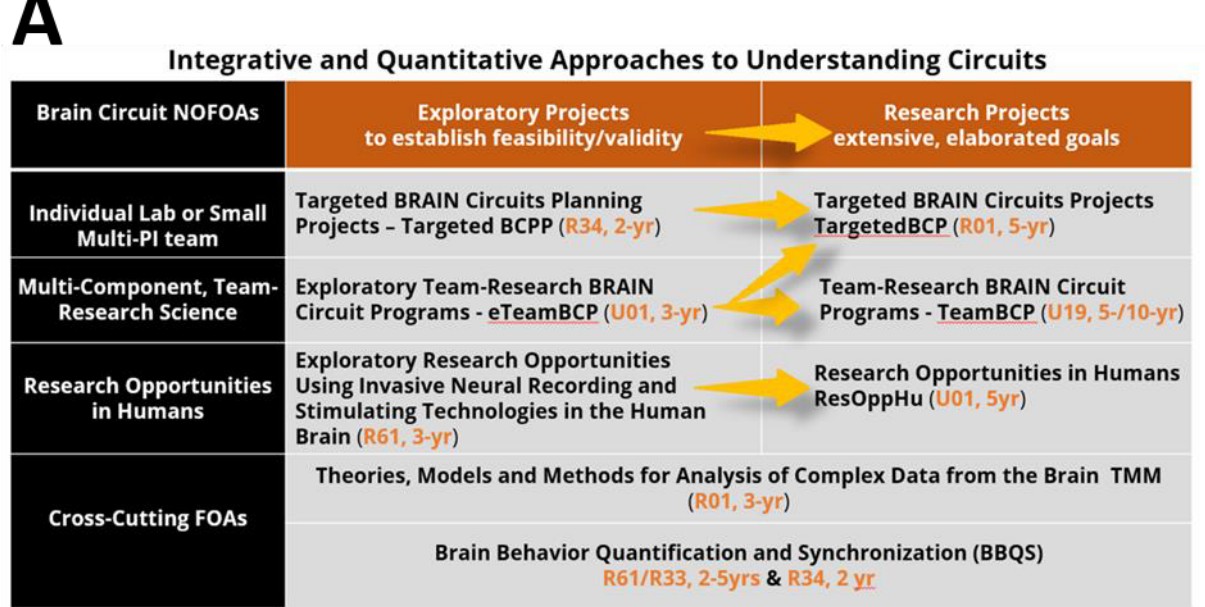

**Figure 1.** Ecosystem of funding mechanisms, awards, and expenditures. (**A**) Ecosystem of funding mechanisms. Exploratory versions of each research track are designed to provide an enabling step, where needed, that goes through peer review to larger, more elaborated awards. (**B**) Tabulation of number of awards and total spending for 2014–2023, not including approved out-years of funds yet to be expended.

emphasis to develop and disseminate computational tools in the form of novel Theories, mechanistic Models, or mathematical Methods (TMM, 3-year award). Parameters built into the program as requirements included (1) end user evaluation of the proposed tools, (2) experimental studies limited to model parameter estimation and/or validity testing of the tools being delivered. In order to focus

the applications on mechanistic understanding at the level of dynamic circuits, after the first cycle of applications, an additional condition specified that products of the research must include knowledge at cellular and sub-second temporal resolution. The TMM 10-year award commitment to computational NS was 7% of the total expenditures at $65M.

The popularity of the top tool products can be quantified by GitHub stars, which indicate that a user likes or finds a software tool useful. *Supplementary file 2A* lists the top TMM awards, categorized as either a Theory, Model, or Method, that have the most repositories. Topics with high popularity include calcium imaging methods, multiple encoding/decoding tools, multiple model-building exercises, and sophisticated statistical methods. The popularity of the top repositories from the TMM program can be quantified by GitHub stars and watchers (*Supplementary file 2B*). These two metrics can be reflective of active repository developers and contributors, which include multiple calcium imaging codes and a matrix factorization tool for imaging analysis.

In the interest of improving measures of high-content, high-temporal resolution of behavior to match that available for neural activity, a novel research track in Basic Behavioral Quantification and Synchronization (BBQS) was initiated in FY2023. There are staged funding tracts specifically for human behavior and for organismal behavior. The human clinical tract employs an administrative review for progression from early to elaborated stages. Whereas, the organismal tract employs a peer-reviewed progression from a 2 year enabling stage to elaborated 5-year stages for non-human and comparative human/non-human studies. These programs are expected to better develop and engage dynamic behavioral and environmental assays integrated with dynamic measure of the neural systems of study. As a newly launched funding program, it is now premature to assess overall impacts for these programs in this report.

## 10-Year impact evaluation

Now 10 years since conceiving this landmark initiative to better understand how the brain works, this report analyzes the impact of this functional neuroscience ecosystem as funding experiments in the domains of systems and integrative neuroscience, and computational neuroscience, with an eye on comparison with other funding models and best practices for how best to support fundamental, investigative neuroscience.

*Figure 2* illustrates the annual NIH research investments committed in investigative neuroscience by year for the 10 years before and after the launch of the BRAIN Initiative. Using the NIH RCDC Terms coding (Research, Condition, and Disease Categorization), which is a computerized reporting system that categorizes grants into public spending areas, we found that 'Neurosciences Research'

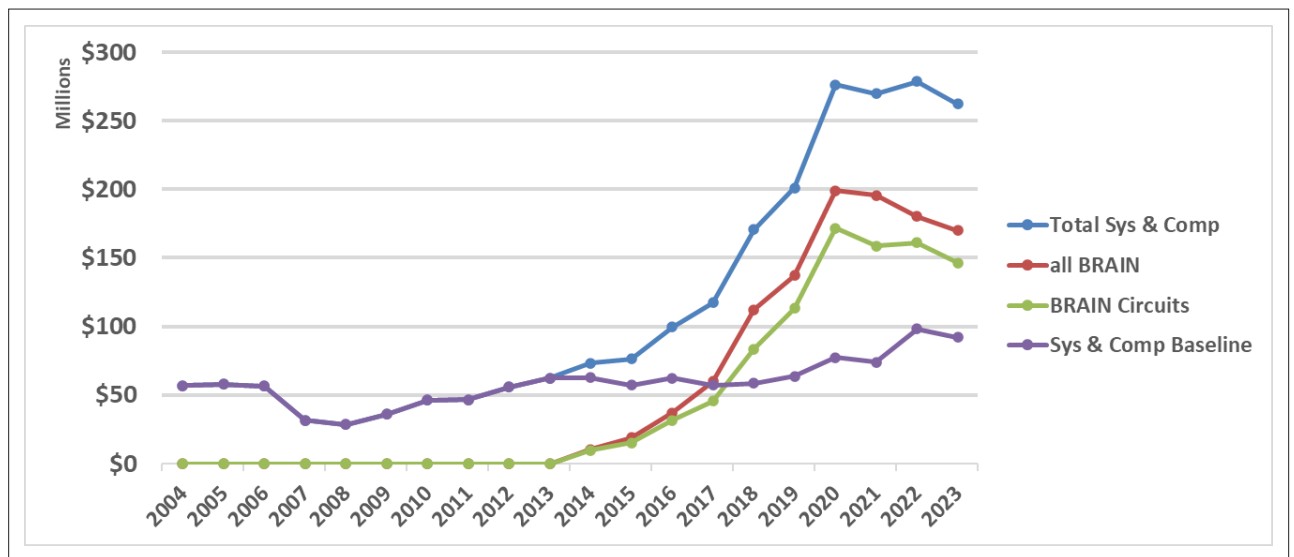

**Figure 2.** NIH expenditures in systems and computational neuroscience for 10 years before and after launch of the BRAIN Initiative. Baseline is calculated from NIH RCDC (Research, Condition, and Disease Categorization) search of 'neurosciences research' [OR] 'computational neuroscience' (Sys & Comp NS), excluding BRAIN Awards. BRAIN Initiative awards are tallied as all Sys & Comp NS within BRAIN and all awards within the BRAIN Circuits funding announcements.

best captured the BRAIN Circuits Program circuit-based research approaches with few false negatives. Thus, we curated a timeline of NIH baseline funding of neuroscience research similar to the BRAIN Circuits Program portfolio using the RCDC terms of the NIH RePORTER knowledge base for 'Neurosciences Research' [OR] 'Computational Neuroscience' (Sys&CompNS2004_2009, Sys&CompNS2010_2015, Sys&CompNS2016_2019, Sys&CompNS2020_2023, due to volume limitations of RePORTER the download is offered as four downloads by date ranges), excluding BRAIN Initiative awards. In order to compare this comprehensive list of NIH awards that includes clinical trials and other nonstandard funding mechanisms to the investigative, fundamental systems and computational neuroscience represented in the BRAIN Circuits Program, the download list was curated to match the NIH funding mechanisms of the BRAIN Circuits portfolio (see METHODOLOGY). We note here that this curated list does include cellular/molecular/developmental neuroscience and disease-specific, basic research, whereas the BRAIN Circuits Program is explicitly disease-agnostic, systems neuroscience. This annualized baseline was tabulated from sums of Awarded Total Costs by fiscal year, and the awards were committed using annual appropriations that include new and competing renewal awards and noncompeting renewals. Unawarded commitments to pending out-years of funding are not included.

Similarly, all BRAIN Initiative awards by year in this categorization, and those issued specifically within the BRAIN Circuits Program, are plotted separately from selection by funding announcements (see METHODOLOGY). The Total Systems and Computational Neuroscience curve is the sum of the neuroscience baseline plus the BRAIN expenditures by year. To manage annual BRAIN expenditures within annualized budget allocations, many of the BRAIN awards were issued as multi-year funding where years 1–3 are funded in the initial award year (including the 3-year TMM) and years 4–5 are awarded after an administrative review. The BRAIN curves rise for fiscal years 2014–2020, including both standard annual awards plus the multi-year budgets for 3 years, and flatten for fiscal years 2021–2023 due to unawarded, noncompeting continuations.

While the NIH baseline funding in this definition of neuroscience research grows relatively steadily by an average of 4.2% per year, the investments from the BRAIN Circuits Program climb by an additional average of 12.5% per year. This creates a more than fourfold accumulation of investment. Since the NIH baseline includes both basic and disease-centered research and neuroscience research outside the heksor concept, the BRAIN Circuits Program increment in funding for basic, disease-agnostic discovery in systems and computational neuroscience is substantially higher than 4×.

*Table 1* reports bibliometric productivity measures for each research track across all awards within each funding mechanism, including the median publications per award, the median citations per award, and the median Relative Citation Ratio (RCR) per publication. For comparison to the BRAIN Circuits Program awards, we include bibliometrics for the 127 non-BRAIN 'Parent' R01 or investigator-initiated research project awards from the FY2014–2023 Systems and Computational baseline (*Figure 2*), linked to 2095 research articles. Publications are a traditional measure of research output by peer review, whereas the rates of citations are a measure of influence to knowledge progress by subsequent reference. The RCR is a normalized index of impact (*Hutchins et al., 2016*), which is a measure of citation relative to the topic neighborhood or subject areas related to the reference article in its co-citation network for the year of publication. An RCR of 1.0 represents a median citation count relative to that

**Table 1.** Bibliometric and fiscal performance measures for each funding mechanism.

| | Median Pubs per Award | Median Cites per Award | Median RCR per Pub | Median Pubs per $1M | Median Cites per $1M |
|---|---|---|---|---|---|
| eROH | 11 | 212 | 2.5 | 5 | 62 |
| ROH | 6 | 73 | 1.9 | 1 | 21 |
| TargetBCPP | 1 | 2 | 1.3 | 1 | 1 |
| TargetBCP | 4 | 48 | 1.7 | 2 | 14 |
| eTeamBCP | 9 | 132 | 1.5 | 3 | 38 |
| TeamBCP | 25 | 669 | 2.6 | 2 | 195 |
| TMM | 9 | 79 | 1.2 | 7 | 23 |
| Parent R01 | 9 | 191 | 1.1 | 5 | 117 |

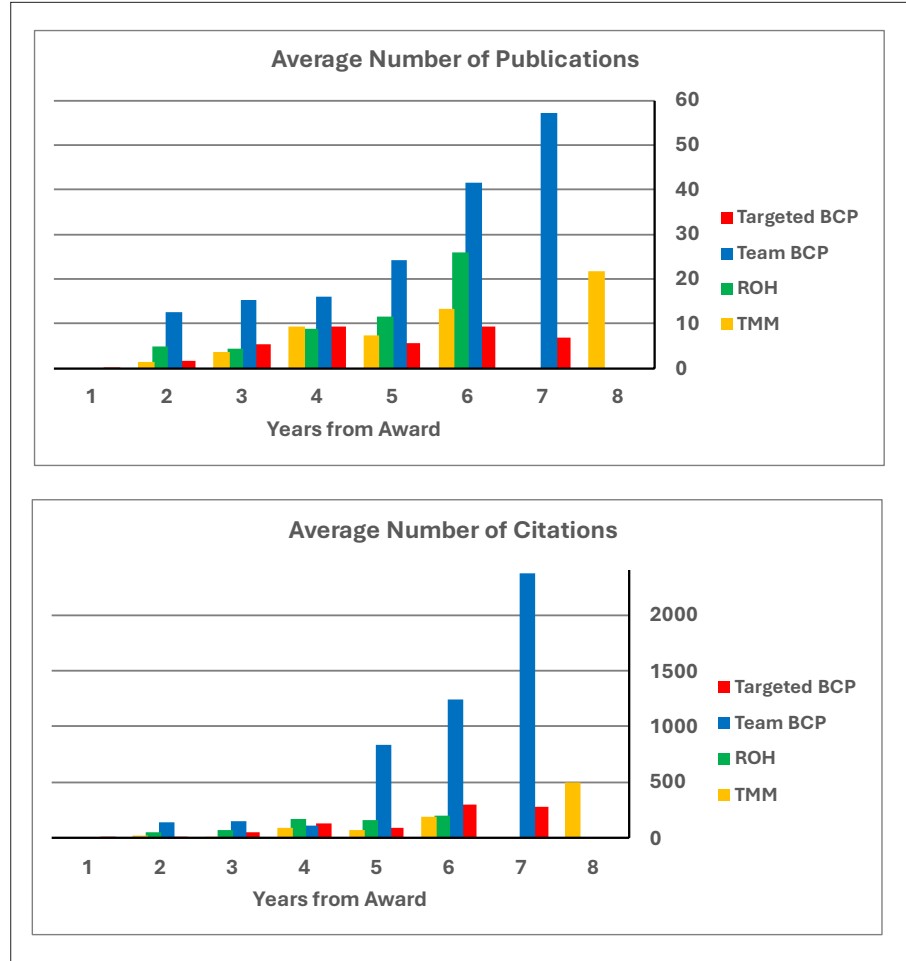

**Figure 3.** Accumulating performance measures (publications and citations) for awards in the elaborated version of each funding tract by year from award.

year and topic neighborhood. While this makes the RCR normalization dependent on the topic neighborhood selection, the methodology of the topic normalization is described in *Hutchins et al., 2016*, and the topic neighborhood for each RCR calculation is available as part of the report.

Comparing the 5-year Parent R01s to the BRAIN 5-year awards (ROH, TargetBCP, and TeamBCP) reveals a much higher publication rate for the TeamBCP and similar publication rates for the other 5-year programs. Interestingly, the exploratory version of the ROH program had notably higher citation rates than the elaborated 5-year ROH awards. In general, the BRAIN awards had fewer median citations than the Parent R01s, while the TeamBCP had substantially more citation rates than any of the other programs. In general, the BRAIN program awards had higher median RCR per publication than the Parent R01 awards. The range of average RCR for the top 5 awards within non-exploratory tracks was TargetedBCP, 5.5–9.3; TeamBCP, 2.5–5.5; ROH, 3.8–5.1; TMM, 2.2–5.7, respectively. Budget efficiency measures (publications and citations per $1M of award) were highest for the TeamBCP, TMM, the Parent R01, and the eROH programs.

*Figure 3* reports successive years from initial award for each of the full-scale programs for each research track (TargetBCP, TeamBCP, ROH, and TMM). All of the programs accumulated publications and citations at a healthy rate of return, especially for the team-research BCP program. By these publication and citation measures, all these systems neuroscience programs featured impressive impacts.

As a measure of the effectiveness of the peer review advancement from exploratory to elaborated funding, for each of the tracks, we calculated the rate of advancement for the exploratory versions to the intended, larger-scale awards (see exemplar methodology below for the TeamBCP funding tract). The BRAIN R34 tract to BRAIN R01 was 13%, reflecting a low-cost, 'high-risk/high-reward' path

to elaborated BRAIN programs. The advancement rate for the exploratory TeamBCP awards was a healthy 59%, and the rate of successful advancement of the exploratory ROH awards was an impressive 73%. When evaluating the advancement rate to any/all subsequent NIH awards, including other ICs and BRAIN, the respective successes rose to 30%, 76%, and 73%.

We similarly calculated the transition successes for other comparable NIH programs for the same range of fiscal years as the BRAIN funding mechanisms. The all-NIH advancement rate for:

- the neuroscience NIH Parent R21, as a comparable pilot program, was 47%;
- the competing renewals for neuroscience NIH Parent R01, as a comparable project continuation rate, was 13%;
- the NIH Director's Transformative Research award R01 program, as a comparable high-risk/high-reward program, was 33%;
- the CRCNS program, as a comparable combination of experimental and computational approaches, was 95%.

Of course, such transition success to subsequent awards is an incomplete measure of impact, as are bibliometric measures that are objectively quantitative but rely heavily on traditional academic publication/citation metrics. For example, one-shot, tool-building exercises might spawn large numbers of subsequent use in the research community but not advancement to a related new award or broad citation, which is particularly characteristic for the BRAIN exploratory TargetedBCPP and the TMM tool-building awards.

Similarly, as a snapshot of public impact, we turned to Altmetric scores within each research tract in *Supplementary file 3*, which are a measure designed to identify how much media attention a research publication has received. The high popularity topics for the large and small Brain Circuits awards include interesting behaviors, such as sleep/wake, maternal behaviors, and odor pleasantness; understanding structural details of the brain; SARS-CoV-2 and anosmia (*Brann et al., 2020*); the discovery of possible fourth meningeal membrane in the brain that helps protect and bathe the brain (*Møllgård et al., 2023*); and elucidation of the neural circuitry underlying maternal responses to infant cries (*Valtcheva et al., 2023*). Top topics for the functional human neuroscience centered around neural decoding of speech/language and neural encoding for speech prostheses. The top topic of public interest for the TMM program involved identifying anxiety activity in mnemonic and

**Table 2.** We report the total number, cost, and number of publications per funding stream for FY2014 through 2023.

| | Team-research | |
| --- | --- | --- |
| | **eTeamBCP (U01)** | **TeamBCP (U19)** |
| Number awards | 40 | 22 |
| Total cost | $118.7M | $280.7M |
| Number publications * | 366 | 618 |
| Median annual total cost per award | $1.0M | $2.9 |
| | **Small project** | |
| | **TargetBCPP (R34)** | **TargetBCP (R01)** |
| Number awards | 60 | 137 |
| Total cost | $36.4M | $312.2M |
| Number publications* | 99 | 624 |
| Median annual total cost per award | $0.3M | $0.6M |

Note this data is frozen based in 2023 and only includes publications published between 2014 and 2023. Please note that some projects are completed while others are not, which adds some variability to publications and the amount of money that has gone out the door (including if projects are multiyear-funded). Finally, when assigning publications to grants, NIH does not typically differentiate between new awards (type 1) and competing renewals (type 2), so for the purposes of publication calculations, type 2s are collapsed into one single project. They are counted separately for the number of awards.

*Papers published from 2014 to 2023 only, some papers may be double counted between programs if more than one program contributed to the publication, but is not double counted within a program if more than one award contributed (i.e. if an eTeamBCP and a TeamBCP both are cited in one publication, the publication is counted under both programs; if two different TargetBCPP awards contribute to one publication, the publication is counted once under the program).

brain hormonal axes. It is noteworthy how some of these projects awarded for their merit in fundamental neuroscience research offer examples of manuscripts that attest to early translational impact, as indicated by asterisks.

### The NIH BRAIN Initiative's Experiment in Team Research (TeamBCP)

An early innovation of the BRAIN Initiative's experiment was to support team-research at a scale commensurate with the complexity of a mechanistic understanding of how the brain works as perhaps the most complex machine in the universe (*David et al., 2020*). Here, specific progress and impact of the program are presented to better understand our funding efforts in large-scale, team-research neuroscience.

*Table 2* summarizes some statistics of the BRAIN team-research programs, compared to the BRAIN small-project circuits programs. The team-approach program was initially issued in 2014 as a 3-year funding opportunity, Integrated Approaches to Understanding Circuit Function in the Nervous System.

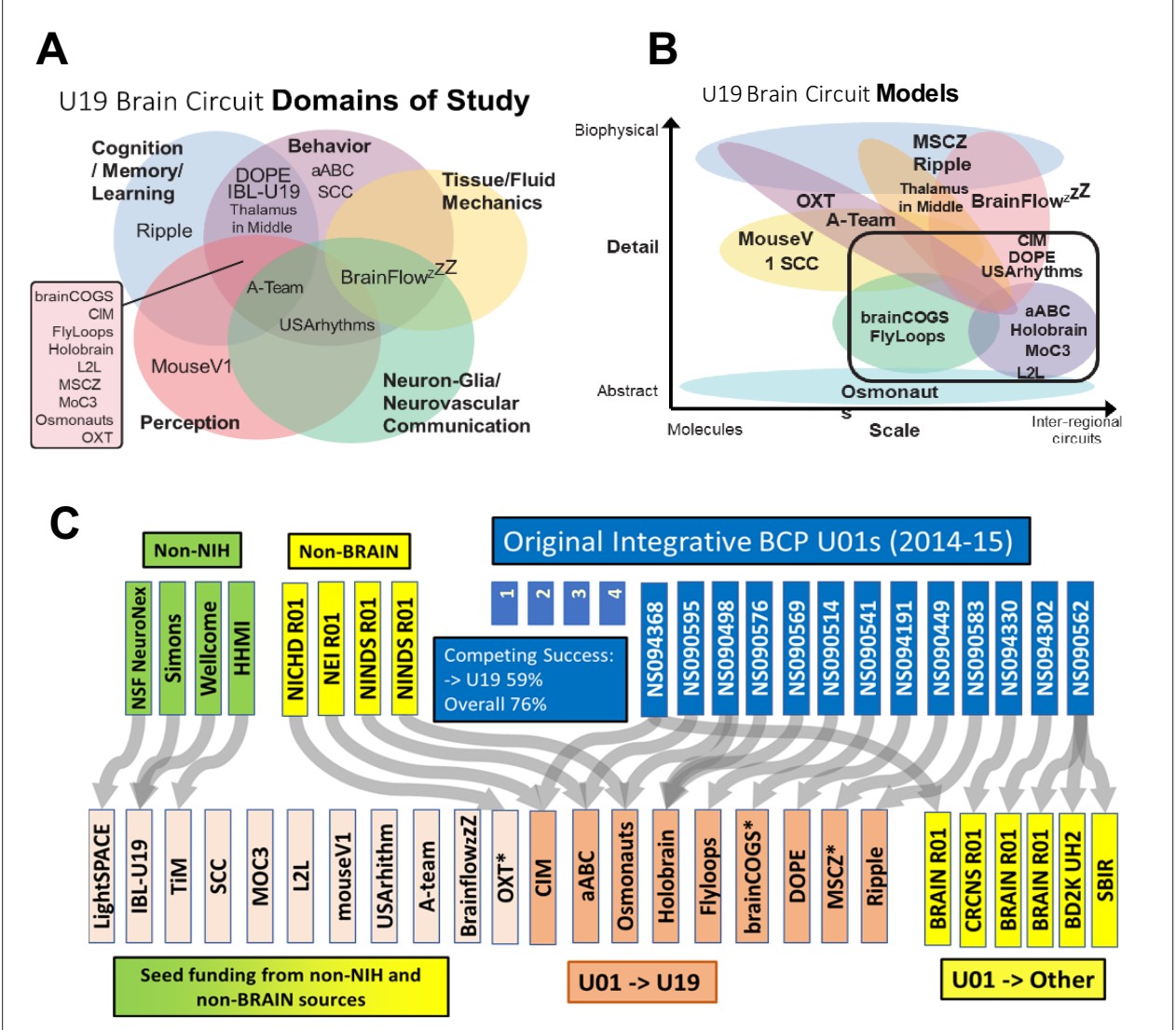

**Figure 4.** The U19 awards are designated according to a branding name to represent the topic of study. Osmonauts (Cracking the Olfactory Code); MouseV1 (Mouse Visual Cortex); Sensation (Coding, Sensation, Behavior); SCC (Spinal Control Circuits); MoC3 (Motor Control Circuits, Computation); MSCZ (Multiscale Circuits of Zebrafish); FlyLoops (Feedback Loops of Flies); aABC (almost Anything But Cortex), DOPE (Dopamine); L2L (Learning to Learn); Ripple (Hippocampal Ripple-Related Episodic Memory); brainCOGS (Brain Circuits of Cognitive Systems); OXT (Oxytocin Group); LightSPACE (All optical read/write in the brain); IBL-U19 International Brain Laboratory; TIM (Thalamus In the Middle); CIM (Causality in Motion); BrainflowzZZ (Cerebral fluid flow and sleep); A-Team (Astrocyte modulation); USARhythms (Neurovascular rhythms). * Advanced to competing renewal as of 2023.

In 2017, the program evolved into the larger-scale, multicomponent TeamBCP funding opportunity of 5 years, and in 2018 the original 3-year program was repurposed as *Exploratory Team-Research BRAIN Circuit Programs – eTeamBCP*, an exploratory stage to enable advancement to the larger, more elaborated 5-year TeamBCP. For FY2014–2023, the total expenditures of the 3-year eTeamBCP, including the original 'Integrated Approaches', were $118.7M; the total expenditures for the TeamBCP were $280.7M (*Table 2*). Note that these expenditures in BRAIN team-science approaches were balanced with parallel, small-project awards, the 3/5-year Targeted BRAIN280 exploratory and elaborated programs at $348M (*Table 1*). In FY2023, the expenditures in this team-science approach were $129.4M, which was 19% of the total BRAIN Initiative expenditures.

All TeamBCP and the original Integrative BCP awardees joined two consortia for awardees from all years to explore common challenges and best practices: a leadership consortium and a data science consortium. The leadership consortium produced topic maps of (1) domains of study (*Figure 4A*) and scales of research with respect to conceptual models (*Figure 4B*). As investigator-initiated research topics, the total collection of awards covers a broad portfolio of systems and behaviors. The spectrum of specific research topics can be mined from the award links provided above. The modeling efforts of the collective TeamBCP awards (*David et al., 2020*) span broad scales and levels of mechanistic focus, as estimated by the TeamBCP data science consortium.

To measure the success of the peer-reviewed 'pathway' of competing award from exploratory to elaborated projects and programs, we looked at the 'flow' of the original TeamBCP projects using Sankey plots. The Sankey plot tracks all the original 3-year awards (*Figure 4C*, dark blue). Of 17 in 2014–15, 59% went through successful peer-reviewed, competing award to a 5-year TeamBCP (*Figure 4C*, orange). Of the 20 total TeamBCP, 11 (55%) came from novel sources, including 3 (15%) transitioning from non-NIH funding sources. By including successful competing awards to any NIH mechanism (*Figure 4C*, yellow), the advancement of the original TeamBCP to any competing NIH award is 76%. By comparison, the success rate (successful award including new applications and competing renewals) in 2023 of NIH research project grants (RPG, mostly R21 and R01) was 21.3%. As a comparison to a large, topic-focused neuroscience research center, the NIMH Conte Centers recompeting success rate was 73%, and 'fundamental neuroscience' (disease-agnostic research) IC Program Projects (P01) was 71%, as a comparable multicomponent program.

We conclude the funding strategy of the exploratory awards (eTeamBCP) that proceed to larger, more elaborated projects (TeamBCP) through competing peer review was a highly successful mechanism of continuity for highly meritorious research.

Another measure of team-building success is the confederation of multiple projects to a collective new award. The TeamBCP has five examples (20%) of two or more awards converging into one, mostly from non-BRAIN Initiative sources (four of the five examples).

## Discussion

### Culture change and best practices

In addition to these quantitative productivity measures, the program has facilitated a cultural change of systems neuroscience through these multidisciplinary, team-science approaches (*Miller et al., 2024*). Below, we offer some useful detail of how this team-research program has addressed our original guiding principles for the Brain Circuit Programs (*David et al., 2020*).

### A culture of collaboration

We have supported projects that are approaching complex questions at multiple scales, through bridging of disciplines, and across species. This combining of disciplines is reflected, in part, by a demographic survey from the 2022 BRAIN Initiative Investigator's Meeting. This survey of meeting attendees found an emphasis on integration of experimentalists and theorists (Statistics, Computational Biology, Computer Science; 31% of 520 respondents), and research spanning different scales and species (see below). We have also brought in tool developers, usually from physics and/or engineering backgrounds (42%), as well as neuroethicist expertise (5%). All of the TeamBCP awards that include humans in the NIH definition of Clinical Trials (L2L, Osmonauts) had neuroethics components built into peer review. In addition, the topics supported have expanded to include the impact of non-neuronal components on brain circuit dynamics (A-Team, USARhythms, BrainflowzZZ).

## Consortia-level collaboration

The TeamBCP consortia specifically are charged with taking on community interests larger than the individual awards. Despite vast differences in research topics and methods/approaches, the consortia have built a research community that develops and shares best practices in team management, data science (*Schottdorf et al., 2024*), and has promoted many confederated efforts across awards. Several TeamBCP with similar research topics exchange data and best practices by joint laboratory meetings (e.g. aABC & SCC, BrainflowzZZ & USARhythms). In 2023, the data science consortium convened an international Digital Atlas Interest Group on common challenges for digital atlas development. Through the consortia, PIs were introduced to recent insights from the field of Science of Team Science, strengthening connections between theoretical and practical considerations of the practice of team science.

## A change in the workforce

The resource cores offer support for staff scientists and a rich training environment that is producing multifaceted trainees. We also note the value of program managers and thoughtful processes/policies as key to effective coordination and fair governance – a point particularly well addressed by the highly standardized International Brain Laboratory. Trainees from team science awards are exposed to an expanded universe of scientific collaborations/interactions and have greater access to resources, mentors, and multidisciplinary expertise available to them (*David et al., 2020*). However, team science is not without challenges (*National Research Council, 2015*), such as programmatic management of disparate disciplines, authorship policies, and credit assignment (*David et al., 2020*). Annual site visits diligently watch for concern with these challenges (see below). TeamBCP trainees have recently established a working group to focus on trainee outcomes. This group is addressing best practices for trainee-related concerns and to optimize trainee team-science research experiences. Additionally, NINDS recently hosted a webinar on perspectives in training in team science. The webinar highlighted the work of a few TeamBCP-supported early career stage scientists and elaborated on the best practices, lessons learned, and lasting impacts of engaging in collaborative research. As well, the data science consortium recognized through their own survey the diverse expertise requirements for creating robust infrastructure and tools for supporting the TeamBCP scientific pipeline from theory to experiment and analysis.

## Spawning new team-science research programs in neuroscience

BRAIN Initiative Team science approaches have inspired innovative non-BRAIN Initiative funding opportunities for team science, notably the NINDS Collaborative Opportunities for Multidisciplinary, Bold, and Innovative Neuroscience (COMBINE) program, RM1 program. Distinct from the NINDS Program Project Grant (P01) and parent R01, this RM1 program draws from experiences with the BRAIN TeamBCP program and emphasizes tight integration across team expertise and approaches.

## Integrated data science and experimental science

The required data science cores were designed to promote flexible and focused solutions to sharing complex data from diverse disciplines, formats, and scales between component laboratories (*Chapuis and Winter, 2024*). The TeamBCP site visits (see below) revealed various and unique data sharing challenges and solutions that were effectively addressed by the specific data science expertise and resources of the data science cores. The dedicated personnel with data science expertise were highly successful and ensured resources to solve complex data science challenges without burdening the productivity efforts of the research staff. We find this 'bottom-up' approach of project-specific data management complements well 'top-down' BRAIN Initiative Data Science and Informatics programs and Data Archives. A self-evaluation by the data science consortium revealed, however, that improvements can be made to adopt more standards for organizing, documenting, processing, evaluating, and sharing data and software. Particularly salient for the early-career scientists was how to recruit and incentivize data science efforts that are not as easily measurable in traditional neuroscience publication and citation.

## Site visits

Annual site visits are arranged for the awarded TeamBCPs to monitor progress and offer mid-award guidance by an External Advisory Board (EAB). The EAB is composed of an NIH or NSF scientific Program Officer with subject matter expertise, a scientist from other team-based BRAIN awards, and an invited scientist at large. The EAB is charged to help the Program Officers assess progress, offer expert constructive critique to the TeamBCP team, explore scientific opportunities, and highlight the achievements of the awardees and trainees. These site visits, which specifically emphasize guided self-evaluation through a Strengths, Weaknesses, Opportunities, and Threats (SWOT) format, have proven to be one of the most valuable factors in the TeamBCP program for both the scientific goals and progress management.

## Enabling cross-species comparative studies

BRAIN supports investigative and tool development studies that take advantage of and/or enable the special abilities and power of diverse species and comparative experimental systems to address specific questions. Species of study among the eTeamBCP and TeamBCP awards include *Drosophila* fruit flies, *Danio rerio* zebrafish, Mus and Rattus rodents, Rhesus and Marmoset monkeys, *Berghia nudebranch*, octopus, voles, bats, and humans. Through these species, these programs have revealed conserved and diverse mechanisms by which the nervous system produces behavior.

## Combining experimental and computational approaches

The extensive cross-scale experimental modeling in *Figure 4B* attests to the power of bringing quantitative approaches to experimental neurobiology. Greater integration of quantitative, predictive modeling into research projects has led to more cross-talk between theoretical modelers and experimentalists and has enhanced collaborations between mathematical and modeling communities (*David et al., 2020*), revealing the importance of a dynamic interaction between theory and experimentation. Like the NSF/NIH CRCNS program, the engagement of special emphasis review panels for the TeamBCP applications effectively recruited well-tuned expertise for the specific, and sometimes complex, integration of computational and experimental methodologies.

## How are we shaping the next generation of scientists?

Based on observations from our TeamBCPs, the BRAIN-awarded trainees, and job advancement from early career investigators, the next generation of scientists is more multifaceted and can comfortably wear different technology hats and speak multiple scientific languages. From an attendee survey of 722 respondents to 2022 BRAIN Initiative Investigators meeting registration, the percentage of primary or secondary fields was highest for Engineering, Psychology or Behavioral Science, Biochem/Molec/Cellular Biology, Physiology or Systems Biology, and Neuroimaging/Radiology (28% to 18% in descending order). This was followed by Computer Science, Computational Biology, Genetics, Statistics (from 11% to 10% in descending order). Also represented was Clinical Science, Physics, Ethics, and Chemistry (from 8% to 3% in descending order). For example, trainees are able to move with ease across the different scales of inquiry, from cellular to systems to quantitative approaches. The data science consortium found through their self-evaluation that theorists and experimentalists are similarly engaged in data science training, though the level of expertise required to enforce standards in research practice remains high (*Schottdorf et al., 2024*). Nonetheless, the culture reflects an openness to innovative and integrative approaches in the neuroscience research workforce.

## Looking to the future

By conceptual design, team-research programs are successful to the extent that they foster collective efforts that could not have happened as effectively as individually funded projects. Traditionally, NIH standard study sections review applications with the mindset of a single lab's success or competition with one another. But with team science, they are reoriented to think in terms of synergy, co-publication, leveraging collective expertise and resources, and collective capacity to achieve the goals they set out to achieve. We believe the rubric of the funding program description and the review criteria drives an ecosystem of team-research that seeks to cross boundaries of interdisciplinary collaboration at scales beyond the limitations of the 'single-principal-investigator (PI)-led projects' common in biology in the past (*Olds, 2016*). The TeamBCP review criteria emphasize synergy as a qualifying element, and the

annual site visits monitor this closely. This report finds that awardees have used the TeamBCP program to engage more adventurous and necessarily collaborative programs of high impact, on topics that transcend interests of all the 'neuro' NIH Institutes. We note that the BRAIN Initiative is part of an effort commissioning NASEM to produce a consensus study about team science and its opportunities and challenges: https://www.nationalacademies.org/our-work/research-and-application-in-team-science.

Following the 10th year of our experiment in supporting team neuroscience, we offer that the BRAIN TeamBCP has been a leading agent for changing the culture of fundamental neuroscience research to consider collective, team efforts of high impact scientifically and socially. The TeamBCP program offers a successful, collective-research approach that complements other collaborative models in neuroscience (*David et al., 2020*). We encourage that the TeamBCP approach continues to be an enduring model of bold and effective team-research approaches in fundamental neuroscience discovery at NIH. Useful guidance in how to advance academic implementation of team-research approaches can be found in the EMBO reports white paper (*van Helden et al., 2024*).

## Conclusions

The NIH BRAIN Initiative's primary goal is to enable neuroscience broadly (*Ngai, 2024*). Its impact is recognized across the NIH neuroscience IC priorities (see *BRAIN at 10* messages of the 10 neuroscience Institute Directors in The BRAIN Blog). We offer that the BRAIN Initiative Circuits Program has substantially advanced and changed the landscape of research funding in fundamental, pre-translational discovery in systems and computational neuroscience. A qualitative testament to this conclusion is reflected in an informal poll of IC-contributing Program Directors that work within the BRAIN Circuits Program Team (*Supplementary file 4*). For objective measure, the raw and normalized citation indices offer impressive measures of scientific impact across all the funding tracts, and particularly high performance by the team-research TeamBCP program (*Table 1* and *Figure 3*).

In summary, the BRAIN Initiative has invested approximately $1B into fundamental, quantitative, systems neuroscience over 10 years (*Figure 1B*), reaching more than fourfold increase in the rate of funding in basic experimental neuroscience compared to the 10 years prior to the BRAIN Initiative (*Table 1*). Another important feature of the BRAIN Circuits Program was that a strategy of peer-reviewed, staged funding from exploratory/pilot projects to complex, hypothesis-driven research of high merit was highly effective. By measures of publication and citation rates, the BRAIN Circuits Program has produced fiscally efficient, high-impact discovery in basic neuroscience research across a broad spectrum of research topics in behavioral, systems, and computational neuroscience.

## Materials and methods
### Database of awards

In order to have a baseline of system neuroscience awards similar to the Systems Neuroscience [OR] Computational Neuroscience portfolio, data downloads were curated to include research mechanisms similar to the BRAIN Initiative mechanisms of award:

- non-BRAIN R21/U21, R34/35/36/37, R61 to match the BRAIN Circuits Program exploratory projects (R21/U21, R34, 3-yr U01, R61);
- non-BRAIN R01, R15, DP1, DP2, DP5, U01, RF1/UF1 to match the BRAIN Circuits Program R01 and RF1;
- non-BRAIN U19, P01, P50/56, RM1 to match the multicomponent, research-center scale of the BRAIN Circuits Program U19s.

### Budgetary and grant data

BCP program budget and numbers of applications (FY2014–2023) were pulled from NIH FASTR and publications from iCITE. Number of publications, publication citations, median citations, and Relative Citation Rates were extracted from iCITE and custom Python scripts by the BRAIN Office of Budget.

## BRAIN Circuits Program funding announcements for Figure 2

| eROH | ROH | eTargetBCP | TargetBCP | eTeamBCP | TeamBCP | TMM |
|---|---|---|---|---|---|---|
| RFA-NS-16–008 | RFA-NS-19–001 | RFA-NS-17–015 | RFA-NS-17–014 | RFA-NS-14–009 | RFA-NS-17–018 | RFA-EB-15–006 |
| RFA-NS-17–019 | RFA-NS-22–041 | RFA-NS-18–014 | RFA-NS-18–009 | RFA-NS-15–005 | RFA-NS-19–003 | RFA-EB-17–005 |
| RFA-NS-18–010 | | | RFA-NS-21–014 | RFA-NS-18–030 | RFA-NS-18–008 | RFA-NS-22–040 | RFA-EB-20–002 |
| | | RFA-NS-22–027 | RFA-NS-21–013 | RFA-NS-18–009 | | |
| | | | RFA-NS-22–026 | RFA-NS-20–029 | | |
| | | | | RFA-NS-22–028 | | |

### Landscape maps

All of the notices of funding opportunities from the Brain Circuits Portfolio (integrative and quantitative neuroscience) were queried in the NIH internal iSEARCH 3.0 portfolio analysis platform to yield all of the unique awards from 2014 to 2023. The titles and abstracts from the resulting competing awards were input into a Python script using Matplotlib library and Seaborn interface in order to derive the cluster visualizations from words that appeared frequently. The Python script was adapted from https://amueller.github.io/word_cloud/ and can be found here: https://github.com/fbader2/wordcloud (copy archived at *Bader, 2025*).

### Continuity analysis

The BRAIN Circuits Program portfolio of awarded grants was downloaded using an NIH-internal database search engine (Query View Report [QVR]) that can recapitulate the public, awarded grant information available from NIH RePORTER as linked in this report, including the keyword fingerprint field (Matchmaker). For each PI for a project, grants across the NIH that shared similarity in scientific content to the original project (derived from the similarity fingerprint) were identified. Unawarded, non-research-related mechanisms and nonbasic science-related grants were filtered out. A manual review of the title and abstract was used to confirm that the selected project was a continuation of the original grant's work by the original (Multi-)PI.

Continuity analyses were constructed for each of our BRAIN Circuits Programs with the general convention of smaller, exploratory awards leading to larger grants: U01→U19, R34→R01/U01, and eROH→full ROH. The 'downstream' projects that resulted from the original project included both BRAIN and non-BRAIN grants.

Comparison groups were determined by examining non-BRAIN NIH initiatives that had similar goals and funding/administrative structures as the BRAIN announcements. The same set of processes/business rules for continuity plotting was applied.

Success rate calculations within a funding track for the continuity analyses were performed using the following formulas: (Number of similar downstream awards)/(Number of original awards). Note that if multiple original awards led to a similar downstream award, that batch of awards was only counted once in the numerator, and that a given BRAIN award could spin off multiple, successful, downstream awards.

### GitHub data

Customized scripts using all of BRAIN Circuits Program application IDs were written to extract repositories from annual NIH Research Performance Progress Reports in conjunction with the number of stars, watchers, and forks associated with each repository. (Note: This code was developed as part of a larger NIH project which accesses privacy-protected information in privileged NIH systems and, thus, is not reproducible publicly. Queries about this analysis can be directed to the authors.)

# Additional information

## Funding

| Funder | Grant reference number | Author |
|---|---|---|
| National Institutes of Health | BRAIN Initiative | Farah Bader<br>Clayton Bingham<br>Karen K David<br>Hermon Gebrehiwet<br>Crystal L Lantz<br>Grace CY Peng<br>Mauricio Rangel-Gomez<br>James Gnadt |

The funders had no role in study design, data collection and interpretation, or the decision to submit the work for publication.

## Author contributions

Farah Bader, Formal analysis, Writing – review and editing; Clayton Bingham, Formal analysis; Karen K David, Grace CY Peng, Conceptualization, Project administration, Writing – review and editing; Hermon Gebrehiwet, Mauricio Rangel-Gomez, Project administration, Writing – review and editing; Crystal L Lantz, Data curation, Formal analysis, Writing – review and editing; James Gnadt, Conceptualization, Formal analysis, Writing – original draft, Project administration

## Author ORCIDs

Farah Bader ⓘ https://orcid.org/0000-0001-9146-2697
Clayton Bingham ⓘ https://orcid.org/0000-0001-5850-1308
Crystal L Lantz ⓘ https://orcid.org/0000-0002-9763-4725
Grace CY Peng ⓘ https://orcid.org/0000-0003-0169-8124
Mauricio Rangel-Gomez ⓘ https://orcid.org/0000-0002-9079-0346
James Gnadt ⓘ https://orcid.org/0009-0009-1166-5466

Reviewer #1 (Public review): https://doi.org/10.7554/eLife.106136.3.sa1
Reviewer #2 (Public review): https://doi.org/10.7554/eLife.106136.3.sa2
Author response https://doi.org/10.7554/eLife.106136.3.sa3

# Additional files

## Supplementary files

Supplementary file 1. Word Cloud from titles and abstracts of all BRAIN BCP awards. As designed, Behavioral, Systems, and Computational Neuroscience are well represented.

Supplementary file 2. GitHub Data on TMM Awards. (A) TMM projects with the highest number of cumulative GitHub Stars that produced highly popular computational tools in the form of normative theories, predictive models, and computational algorithms. (B) Highest rated GitHub repositories from TMM awards. These highly popular repositories are supported by multiple projects and programs. Snapshot from November 2024.

Supplementary file 3. Publications (PubMed Identifier [PMID]), per research track, from awards that have Altmetric scores above 600. * Manuscripts that attest to early translational impact.

Supplementary file 4. Sample quotations from an informal poll of IC-contributing Program Directors that work within the BRAIN Circuits Program Team.

MDAR checklist

## Data availability

All data generated or analysed for this study are included or linked in the manuscript and supporting files.

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
