## [Editor Report · eLife Assessment]

The authors provide a **convincing** summary of ten years of Brain Initiative funding including the historical development, the specific funding mechanisms, and examples of grants funded and work produced. It is particularly **valuable** at this moment in history, given the cataclysmic changes in the US government structure and function occurring in early 2025.

---

## [Referee Report · Reviewer #1 (Public review)]

This is a convincing description of approximately ten years of funding from the NIH BRAIN initiative. It is of particular value at this moment in history, given the cataclysmic changes in the US government structure and function occurring in early 2025.

The paper contains a fair bit of documentation so that the curious reader can actually parse what this BRAIN program funded. The authors are able to draw on a wealth of real-life experience reviewing, funding, and administering large team projects, and assessing how well they achieve their goals. In revision, the paper has been improved with respect to clarity and by bringing together two separate papers into one stronger piece.

---

## [Referee Report · Reviewer #2 (Public review)]

Summary:

The authors provide an important summary of ten years of Brain Initiative funding including a description of the historical development of the initiative, the specific funding mechanisms utilized, and examples of grants funded and work produced. The authors also conduct analyses of the impact on overall funding in Systems and Computational Neuroscience, the raw and field normalized bibliographic impact of the work, the social media impact of the funded work, and the popularity of some tools developed.

The authors have improved the presentation by integrating the weaker of the two manuscripts with the stronger, by clarifying terminology and by performing additional analyses.

---

## [Author Response]

The following is the authors’ response to the original reviews

**Public Reviews:**

**Reviewer #1 (Public review):**
Summary:In this useful narrative, the authors attempt to capture their experience of the success of team projects for the scientific community.Strengths:The authors are able to draw on a wealth of real-life experience reviewing, funding, and administering large team projects, and assessing how well they achieve their goals.Weaknesses:The utility of the RCR as a measure is questionable. I am not sure if this really makes the case for the success of these projects. The conclusions do not depend on Figure 1.

We respectfully disagree about the utility of the RCR, particularly because it is metric that is normalized by both year and topical area. We have added a more detailed description of how the RCR is calculated on page 6-7. Please note that figure 1 is aimed to highlight the funding opportunities, investments and number of awards associated with small lab (exploratory) versus team (elaborated, mature) research rather than a description of publication metrics.

**Reviewer #2 (Public review):**
Summary:The authors review the history of the team projects within the Brain initiative and analyze their success in progression to additional rounds of funding and their bibliographic impact.Strengths:The history of the team projects and the fact that many had renewed funding and produced impactful papers is well documented.Weaknesses:The core bibliographic and funding impact results have largely been reported in the companion manuscript and so represent "double dipping" I presume the slight disagreement in the number of grants (by one) represents a single grant that was not deemed to address systems/computational neuroscience. The single figure is relatively uninformative. The domains of study are sufficiently large and overlapping that there seems to be little information gained from the graphic and the Sankey plot could be simply summarized by rates of competing success.

While we sincerely appreciate the feedback, we chose to retain these plots on domains and models to provide a sense of the broad spectrum of research topics contained in our TeamBCP awards. Further details on the awards can be derived from the award links provided in the text. Additionally, we retained the Sankey plots because these are a visual depiction of how awards transition from one mechanism to another, evolve in their funding sources, and advance in their research trajectories. The plot is an example of our continuity analysis which is only reported in the text and not visually shown for the remaining BCP programs.

**Recommendations for the authors:**

**Editorial note:**
In the discussion, the reviewers agreed that the present manuscript does not make a sufficient independent contribution and so would be more profitably combined with the companion manuscript. Both reviewers noted that there was not much insight that relied on the single figure. Since neither manuscript is long, and they have overlapping authors (including the same first and last authors), this should not be a difficult merger to achieve.

Thank you for the recommendation to merge. We have combined both manuscripts into one in this version.

**Reviewer #1 (Recommendations for the authors):**
The jargon of the grant programs could be described as a nightmare. Wellcome is spelled wrong.

We have attempted to limit the use of jargon and to define acronyms in this version. We have corrected the spelling of Wellcome.

**Reviewer #2 (Recommendations for the authors):**
I suggest that the two manuscripts be combined into a single paper. Although the other manuscript could stand on its own, this one does not.The idea of culture change surrounding teams is useful but really forms more of a policy- focused opinion piece than a quantitative analysis of funding impact.If the authors insist on keeping these separate, it is critical to remove the team data from the other manuscript.

We have combined both manuscripts and decided to retain the description of culture change but have edited and condensed this section and will use the supplemental report for qualitative assessments.